# Full Mouth Treatment of Early Childhood Caries with Zirconia Dental Crowns: A Case Report

**DOI:** 10.3390/children10030488

**Published:** 2023-03-02

**Authors:** Christina Kanareli, Marine Balazuc-Armbruster, Ioannis A. Tsolakis, Takis Kanarelis, Apostolos I. Tsolakis

**Affiliations:** 1Department of Dentistry, School of Dentistry, University of Reims Champagne-Ardenne, 51100 Reims, France; 2Department of Orthodontics, School of Dentistry, University of Paris V and VII, 75000 Paris, France; 3Department of Orthodontics, School of Dentistry, Aristotle University of Thessaloniki, 541 24 Thessaloniki, Greece; 4Department of Orthodontics, Case Western Reserve University, Cleveland, OH 44106, USA; 5Department of Orthodontics, School of Dentistry, National and Kapodistrian University of Athens, 157 72 Athens, Greece

**Keywords:** zirconia pediatric crown, deciduous tooth, esthetics, general anesthesia

## Abstract

Pediatric dentists should always strive for cosmetic and functional rehabilitation when treating decaying or injured primary teeth. The most popular restoration technique for such teeth is “strip crowns” made of composite materials, but more recently, zirconia and preveneered stainless steel crowns have gained popularity. Moreover, zirconia crowns are usually preferred over stainless steel crowns for aesthetic reasons. The aim of this case report is to present a 4-year-old patient with a high caries risk who underwent a full-mouth pediatric zirconia crown treatment. The operation was performed under general anesthesia. This article describes the restoration of all primary teeth and the follow-up 6 months after the parents’ request to improve the aesthetics. The gingival health and the adaptation of the zirconia crown were evaluated both clinically and radiologically. In this case, the preformed pediatric zirconia crowns provided functional and durable restoration while restoring the natural appearance of the teeth. However, primary dental prevention, including education on oral hygiene and dietary habits, remains the cornerstone in preventing early childhood caries and promoting oral health in pediatric patients. It is important to note that dental intervention under general anesthesia should only be considered as a last resort after all other options have been exhausted, due to the potential risks associated with the use of general anesthesia.

## 1. Introduction

Early childhood caries (ECC) is a very widespread multifactorial disease that constitutes a public health problem in both developing and industrialized countries [1,2]. According to the American Association of Pediatric Dentistry (AAPD), ECC is defined by the presence of one or more carious tooth surfaces (with or without cavities), the absence of one or more deciduous teeth and/or one or more deciduous teeth with fillings due to caries, in children under 6 years of age. It is one of the most common childhood illnesses, affecting 60% to 90% of children worldwide [3]. Even though primary dental prevention, the use of fluoridated toothpaste, and continuing to use fluoride in various forms might prevent tooth decay [4], ECC is still common among kids in underprivileged neighborhoods in both developed and developing nations [5]. Primary dental prevention involves implementing measures to prevent the onset of dental caries, secondary prevention focuses on the early detection and treatment of caries, and tertiary prevention involves managing and restoring teeth affected by advanced caries. Untreated tooth decay affects a child’s growth and development and makes it difficult for them to eat and sleep [6]. According to the International Federation of Dentistry (FDI), ECC is one of the leading causes of absences from school in a number of nations [6]. It can advance quickly, causing discomfort and infection as well as lowering the quality of life for children who have oral health issues [7]. These issues could develop into a severe, perhaps fatal condition [8].

Despite a drop in dental caries in adults, several nations have seen an increase in the prevalence of caries in young children [9]. Dental decay is very common in preschoolers in Southeast Asia. Their dmft (decayed, missing, and filled primary teeth) score, a metric recommended by the World Health Organization (WHO), for their caries experience was 5.1 [10], and their median caries prevalence was 79%. Low-income nations appear to be experiencing a worsening of this issue. For instance, in Cambodia, the frequency of caries among 6-year-old children reached 91%, and their mean dmft score was 7.9 [11]. High caries prevalence (74%) and a significant percentage of untreated decaying teeth (95%) were found in Vietnam [12]. The two nations with the largest populations are China and India. When compared to high-income nations such as the USA (23%) [13], the UK (28%) [14], and other developing nations, the prevalence of caries in preschool-aged children is relatively high at 66% in China [15] and 63% in India [16].

Planning operational interventions for (severe) ECC presents many difficulties for dentists:Primary teeth have a different morphology than permanent teeth because they have a larger pulp chamber and a thinner enamel-dentin layer, which reduces the amount of time it takes for caries to reach the pulp and increases the risk of pain and pulp complications.The child’s future oral health-related quality of life may be compromised by the pain and terror brought on by ECC in emergency dental care.Young children are immature, typically uncooperative (sedation or GA may be necessary), and have a more complex decision-making process because parents are responsible for them.Losing primary teeth too soon can lead to speech impediments, feeding and biting issues, and a delayed eruption of permanent teeth.

The American Association of Pediatric Dentistry (AAPD) and British Society for Pediatric Dentistry (BSPD) guidelines recommend treating primary molars with a preformed pediatric crown when one or two surface caries are extensive [17,18].

The stainless-steel crown (SSC) is the gold standard in pediatric dentistry for the treatment of caries, and this therapeutic option has good clinical and radiological results [19], including longevity over time, with several indications, for deciduous teeth after a pulpotomy/pulpectomy, for teeth with developmental defects or large carious lesions involving multiple surfaces where an amalgam filling is likely to fail, and for fractured teeth [20]. However, the stainless-steel crown (SSC) is often refused by parents for aesthetic reasons. Nowadays, more aesthetic alternatives are available, such as pediatric zirconia crowns [21,22].

Zirconia crowns are a relatively recent issue in pediatric dentistry. Zirconia crowns are an alternative treatment that meets the aesthetic requests of the patients not only in the anterior [23] but also in the posterior sectors. Zirconia is a crystalline dioxide of zirconium, which is sometimes referred to as “ceramic steel.” Zirconia crowns are formed from a single densely sintered block of crystals, which makes them almost three times stronger than porcelain-fused-to-metal crowns and gives them a more translucent appearance [24]. The aesthetic management of deciduous anterior teeth with extensive caries often requires full coronal coverage restoration, which is usually a challenge for pediatric dentists [25]. EZ-pedo developed and produced commercially available pediatric zirconia crowns in 2008. Companies like Nusmile, Kinderkrowns, Cheng Crowns, Signature Crowns, and many others later popularized preformed zirconia crowns [26]. The size, shape, color, and pattern of the retention component varies amongst these prefabricated crowns. Zirconia crowns have several advantages other than aesthetics. They are characterized by the absence of any components that could debond, and possibly by their reduced technique sensitivity. Additionally, because of their highly polished surface, zirconia crowns demonstrate less plaque deposition than those made of other materials. Dental professionals must prepare the teeth to fit the zirconia crowns because they cannot be crimped for adjustment. As a result, primary teeth that are getting these crowns need more preparation than teeth that are getting stainless steel crowns. Therefore, the potential drawbacks of zirconia crowns include the requirement for additional tooth reduction, the inability to crimp or contour the crown, and the cost [27]. 

This article reports the successful treatment of a 4-year-old boy with high individual caries risk, severe dental anxiety and phobia, and aesthetic demand all under general anesthesia by using zirconia dental crowns. 

## 2. Case Presentation

This article describes the restoration of all deciduous teeth in a 4-year-old boy with high individual caries risk under general anesthesia [28,29]. The data collection (patient’s identification) and the constitution of the patients’ medical file (medical assessment) were assessed and revealed an absence of any general pathology. Clinical (Figure 1a–e) and radiological examinations (Figure 2a–d) were also carried out. The patient had previously been seen by five other dental practitioners but had always been extremely anxious during treatment. The use of gas sedation to help calm the child had already been attempted, but unfortunately, it did not work due to buccal respiration, and the child developed a phobia of dental treatment. Given the severity of the child’s anxiety and phobia, it was decided that the best course of action would be to treat the child under general anesthesia. This allowed us to provide the necessary treatment while ensuring that the child remained comfortable and calm throughout the procedure.

A high caries risk was recorded [30] with the following risk factors: absence of twice-daily tooth brushing and absence of use of a fluoride toothpaste [31]. The patient’s eating habits were mainly liquid, soft, or semi-solid food, and specifically four semi-solid or soft meals (breakfast, lunch, snack, and dinner). In addition to that, a sweetened bottle, specifically in the morning and at bedtime, was consumed daily. The patient occasionally complained of discomfort in sector 8 and has been teased at school. 

The pre-selection of an appropriate crown size was estimated prior to tooth preparation by measuring the mesiodistal dimension of each tooth. The total excavation of carious tissue was followed by the pulpal treatment of each tooth. Glass ionomer restorations were made to reconstruct tooth volume (Figure 3a,e,f). A reduction of 1.5 to 2 mm was made occlusally, 0.8 to 1.75 mm was made vestibular and lingually while following the natural contours of the original tooth. The preparation was carefully extended subgingivally by 1–2 mm, removing any gaps to allow a good fit of the preformed pediatric zirconia crown. Finally, the preparations were polished to remove all sharp angles.

The fitting of the preformed pediatric crowns was necessary to confirm the sizes chosen beforehand. The pediatric teeth and crowns were cleaned of all blood residue, and hemostasis was achieved by applying woven compresses. Cementation of pediatric crowns was performed with self-curing glass ionomer cement (Fuji One PLUS, GC) (Figure 3b–d,g,h).

## 3. Result and Follow-Up

The patient still receives follow-ups in a private pediatric odontology practice. Clinical photographs and postoperative radiographs 6 months after the intervention were taken (Figure 4a–e). The suitability of preformed pediatric zirconia crowns and gingival health were evaluated clinically and radiographically (Figure 5a,b). There are no signs of attrition (zirconia versus zirconia), nor are there cracks in the crowns.

In our case, the preparation of the teeth was quite aggressive for the periodontium, which is often the case during the dental preparation of a pediatric zirconia crown. However, 6 months after the intervention, the gingiva healed, and our patient has healthy periodontal tissues (Figure 6).

## 4. Discussion

Motivating our patient for dental care was essential for ensuring the efficacy of treatment, particularly regarding the promotion of oral hygiene practices, such as tooth brushing, utilization of fluoride toothpaste, and adoption of a healthy diet. This was achieved through the provision of information on proper tooth brushing techniques, the importance of fluoride toothpaste, and the detrimental effects of excessive sugar consumption on oral health. Trust and a positive relationship with the patient and his family was crucial in initiating motivation for dental care and was accomplished through the provision of age-appropriate explanations of treatment and involving the family in the treatment planning process. Employing positive reinforcement and reward systems was also used to encourage our patient to cooperate during treatment and maintain good oral hygiene habits at home. However, the patient was still anxious due to previous dental experiences.

The treatment was performed under general anesthesia because placing a pediatric zirconia crown requires not only adequate clinical skills [32] but also the full cooperation of the child. Given the number of preparations to be made, the age of the patient, and the patient’s cooperation, and after having assessed the quality/risk ratio [33], we recommended an operation under general anesthesia. However, the placement of a pediatric zirconia crown in a patient with good cooperation can be considered locally, without resorting to general anesthesia, with or without the use of conscious sedation. When considering dental interventions, general anesthesia should be regarded as a last resort after alternative methods have been deemed unsuitable or have failed to provide adequate patient comfort. The use of general anesthesia carries with it inherent risks, including the potential for adverse reactions and procedural complications. Therefore, an extensive evaluation of the patient’s medical history and individual circumstances is crucial prior to utilizing general anesthesia. In the case of the 4-year-old patient with a high risk of dental caries, the use of gas sedation had already been attempted but was unsuccessful due to the patient’s buccal respiration, which led to the development of dental phobia. As a result, the decision was made to proceed with general anesthesia to ensure the patient’s comfort and safety during the procedure. However, it is essential to note that the use of general anesthesia should not be routine and should only be utilized following careful consideration of the risks and benefits in each individual case.

Following the total excavation of the carious tissues, pulpal treatment was needed due to the extent of the caries, which affected at least two surfaces on each tooth. Biodentine cameral pulpotomies were performed on all deciduous teeth [34]. Regarding the mandibular incisor sector, composites could be considered. However, several parameters were considered and influenced our therapeutic approach: due to a lack of patient cooperation, a loss of a composite or the occurrence of infectious incidents would require re-intervention under general anesthesia. Thus, our clinical approach was more invasive, and zirconia crowns encompassing the entire dental surface were used, reducing the risk of secondary caries.

Despite the fact that the stainless-steel crown (SSC) is the gold standard in pediatric dentistry for the treatment of extensive caries, the choice of pediatric zirconia crowns was based on strong aesthetic demand [35,36]. Stainless steel crowns (SSC) need less invasive preparation compared to pediatric zirconia crowns. However, pediatric zirconia crowns are thicker and less flexible compared to preformed metal crowns, and preparations without any sharp or subgingival angles are required [27].

In our case, tooth preparation was quite aggressive for the periodontium, which is often the case during dental preparation for a pediatric zirconia crown [37,38]. However, the gingiva healed, and the long follow-up term also indicated healthy periodontal tissues. However, the healing and the absence of gingival inflammation also depends on the oral hygiene of the patient.

The adjustment of the occlusion was complicated during the intervention because all of the teeth were prepared and reduced thus, any occlusal reference was maintained. In single preparation of a zirconia pediatric crown, the reference point usually used to avoid an overbite or an open bite is the marginal ridge of the adjacent teeth. It is necessary to perform single restorations first before performing consecutive restorations because there is a learning curve for the placement of crowns [19]. Even though the preformed zirconia pediatric crowns have been tried in before final cementation, we observed a bilateral posterior open bite because it was not possible to check the occlusion either statically or dynamically because the operation was performed under general anesthesia. However, the result is quite satisfactory both aesthetically and functionally, as the patient no longer complains of pain and has resumed a solid diet. Our aesthetic objective was accomplished because we responded to the request of the parents and our patient [39].

The treatment was carried out in a private dental clinic, and the cost of the treatment was covered by the patient’s family. In terms of long-term follow-up, the patient will continue to be monitored and screened at the dental clinic until the exfoliation of the deciduous teeth. It is important to note that the use of general anesthesia in this case was a last resort due to the patient’s severe anxiety and phobia towards dental treatment. Efforts should be made to address and manage the psychological and emotional aspects of pediatric dental care in addition to addressing the medical and clinical aspects. Additionally, cost can be a significant barrier for some families, and therefore, alternative options such as community clinics or government-funded programs should also be considered and made available. In terms of future works and recommendations, it is important to address and manage the psychological and emotional aspects of pediatric dental care. It is also important to investigate and implement cost-effective treatment options and to increase patient and family engagement in oral hygiene practices. Furthermore, addressing cultural and linguistic barriers in patient education and communication should be considered. Continuously monitoring and evaluating the effectiveness of the approach through long-term follow-up is also important.

This case can be considered valuable for the discipline of pediatric dentistry, as it describes the successful treatment of a 4-year-old boy with high individual caries risk, severe dental anxiety and phobia, and an aesthetic demand all under general anesthesia, and there are no similar cases available in the literature. The case highlights the importance of addressing and managing high caries risk, severe dental anxiety and phobia, and aesthetic demands in children, particularly in cases where traditional methods of behavior management and sedation have proven ineffective. The approach allowed for the necessary treatment to be performed while ensuring the child’s comfort and well-being, addressing the underlying factors contributing to the child’s high caries risk, such as poor oral hygiene habits and a diet high in sugary foods, and meeting the aesthetic demands of the child.

## 5. Conclusions

It is important to note that dental intervention under general anesthesia should only be considered as a last resort after all other options have been exhausted, due to the potential risks associated with the use of general anesthesia. Therefore, it is essential to carefully assess the patient’s individual circumstances and medical history before making a decision to proceed with general anesthesia. In the case of the 4-year-old boy with high caries risk, all other methods had been attempted but had failed to alleviate his anxiety and allow for successful treatment. Under these circumstances, the use of general anesthesia was deemed the best option for ensuring the child’s comfort and safety during the procedure. Preformed pediatric zirconia crowns appear to be an aesthetic alternative to the stainless-steel crown (SSC). The 6-month follow-up of our patient, who underwent a full-mouth zirconia treatment, showed very good integration of the pediatric zirconia crowns, both clinically and radiologically. Despite the possibility of considering alternative treatment plans for the same patient, the current approach was adopted given the patient’s poor level of cooperation and the likelihood of future failures with other methods. It is emphasized, however, that primary dental prevention, encompassing appropriate oral hygiene practices and dietary habits, is of utmost importance in preventing dental caries in permanent dentition, and that the management of caries merely addresses the associated symptomatology without tackling the fundamental etiology. A longer and more regular follow-up period should be established until the exfoliation of his primary teeth.

## Figures and Tables

**Figure 1 children-10-00488-f001:**
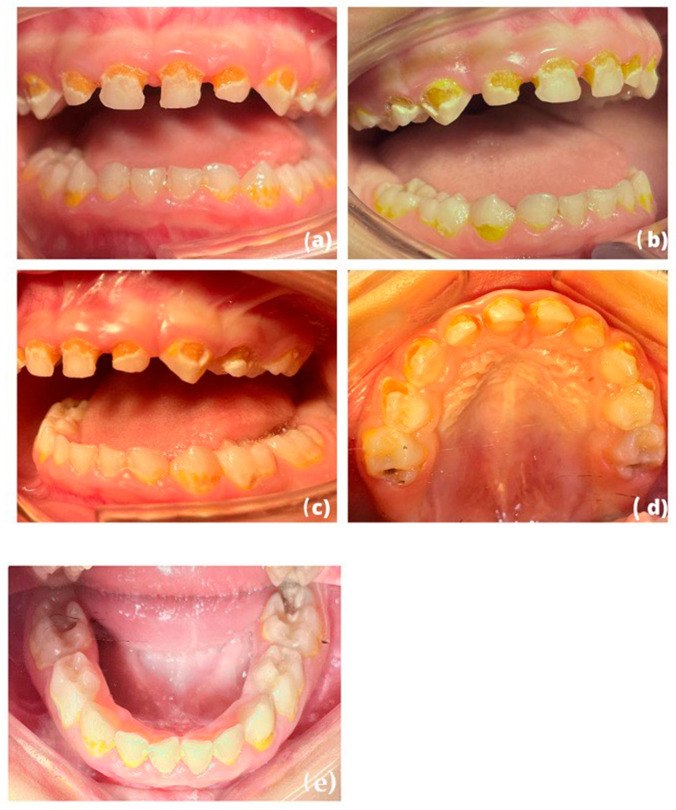
(**a**–**e**): Endobuccal photograph taken during the intervention under general anesthesia (frontal, right, left, maxillary, and mandibular occlusal view) of our 4-year-old patient illustrating the severity of the clinical situation: early childhood caries affecting all of his teeth.

**Figure 2 children-10-00488-f002:**
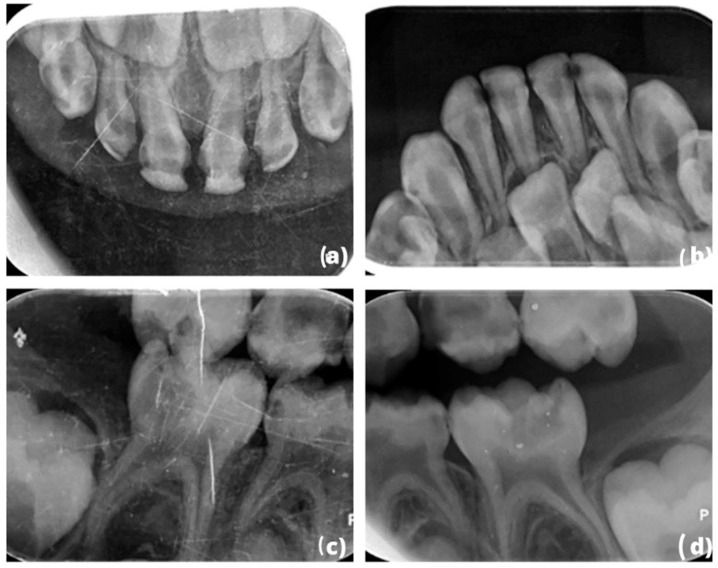
(**a**–**d**): Preoperative x-rays: (**a**,**b**) occlusal x-ray of the maxillary and mandibular situation and (**c**,**d**) bitewing of the right–left side.

**Figure 3 children-10-00488-f003:**
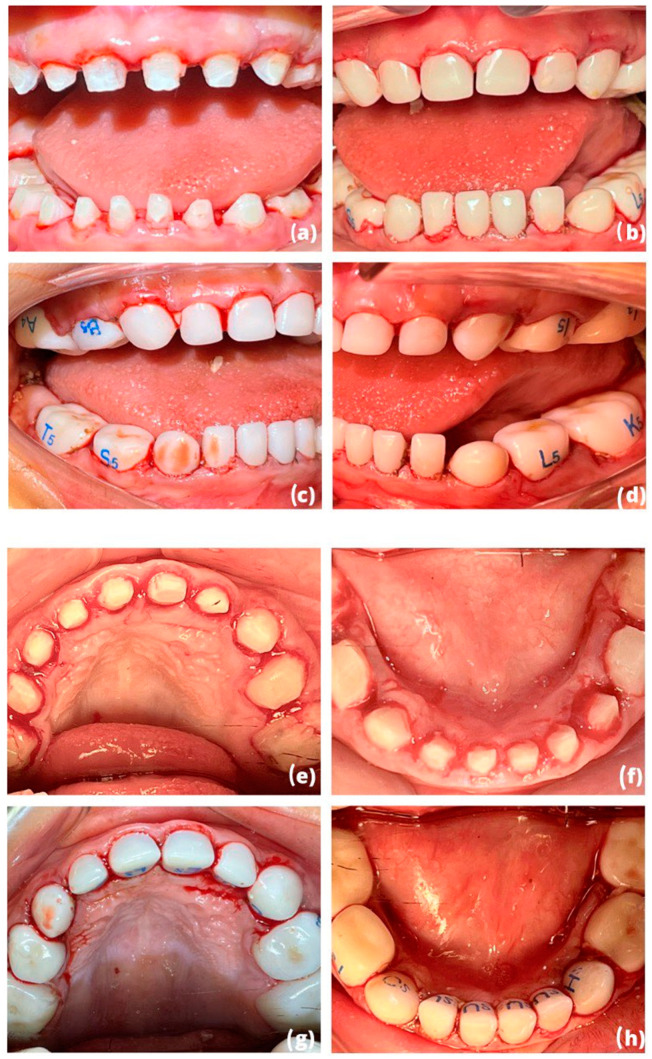
(**a**–**h**): (**a**,**e**,**f**) Intraoperative situation after performing pulpotomy and polishing of our preparations (clinical view); final situation after cementation of preformed pediatric zirconia crowns (**b**–**d**,**g**,**h**).

**Figure 4 children-10-00488-f004:**
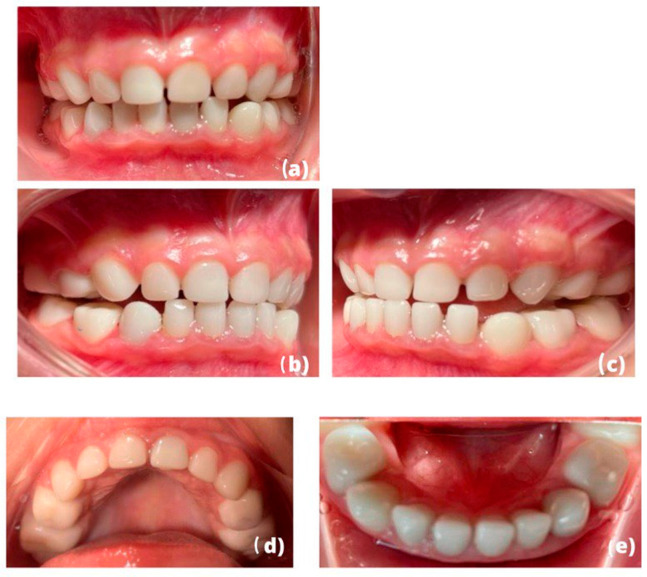
(**a**–**e**). Clinical situation at 6 months post-intervention: (**a**) in occlusion, (**b**,**c**) right and left side view, and (**d**,**e**) upper and lower maxilla.

**Figure 5 children-10-00488-f005:**
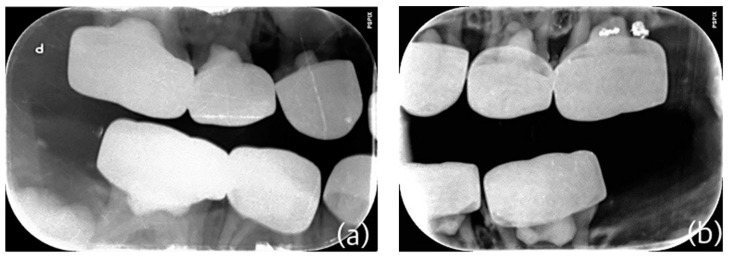
(**a**,**b**). Retro-coronary X-ray 6 months post-operative (right side (**a**) and left side (**b**)).

**Figure 6 children-10-00488-f006:**
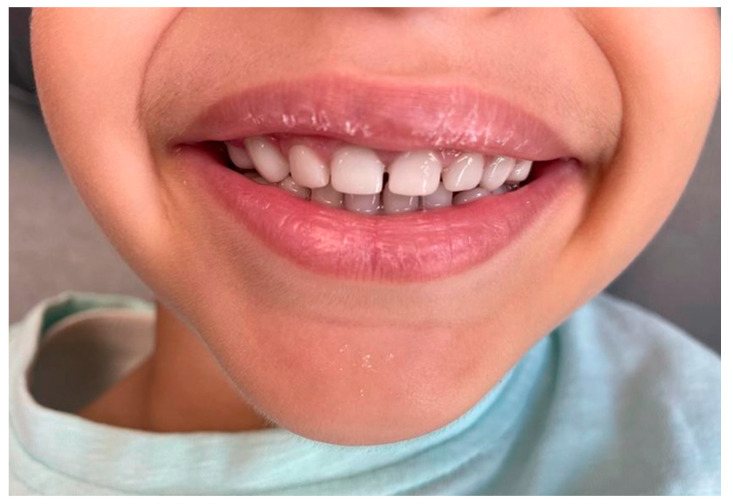
Smile integration at 6 months post-procedure.

## Data Availability

Upon request, the authors can promptly provide any additional data needed.

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
