# Peer review of "Full Mouth Treatment of Early Childhood Caries with Zirconia Dental Crowns: A Case Report"

_children, 2023, doi:10.3390/children10030488_

Round 1

Reviewer 1 Report

In general, the clinical case report is well presented and structured. The description of the case is complete and allows a logical sequence to be followed for its understanding; however, I suggest some recommendations to improve the quality of the manuscript:

·       The introduction section needs to be reordered to improve its logical flow and understanding.

·       In relation to the case, it should be specified why the case can be considered of value for the discipline, what is the main contribution and because it helps the clinician who cares for this type of patient in their clinical.

·       The introduction should be limited only to the problem to be solved; the introduction should be synthesized.

·       One aspect to improve is the description of the figures, they are very good and adequate images; however, only 1, 2 and 3 are cited in the text and the description is very poor. I think that one way to solve this issue is to group and make scenarios like Figure 1A, 1B, 1C, etc.

·       Be specific in what type of pulp treatment was used in each of the teeth.

·       The discussion is practically a continuation of the introduction; must be redone. The report can be of great value because of the little information available on similar cases; however, it should be better addressed and the discussion deepened, describing what the true value of the case is, how clinicians can take advantage of this information, the diagnostic value, the implications, and what can be done at the dental care level.

·       Discussion need more clinical interpretation. The discussion is very descriptive and does not deepen or compare with other treatment alternatives. Individual sentences are presented but not discussed or compared.

·       One aspect to discuss is the cost of treatment and establish in what type of practice this treatment option can be used. In the discussion describe what is expected with a long-term follow-up.

Clarify and deeper the limitations identified and especially the future proposed works and recommendations based on the information generated in the article.

·       Normally, clinical cases have between 10 and 15 references. It is very important to limit yourself to this number and highlight the relevance of the clinical results obtained. For example, in the next sentence, is nor necessary the references “This article describes the restoration of all deciduous teeth in a 4-year-old boy with 96 high individual caries risk under general anesthesia [28,29].” It is important to check all text.

Author Response

Reviewer 1:

In general, the clinical case report is well presented and structured. The description of the case is complete and allows a logical sequence to be followed for its understanding; however, I suggest some recommendations to improve the quality of the manuscript:

  1. The introduction section needs to be reordered to improve its logical flow and understanding.

Thank you for your feedback on the introduction section. We have taken your comment into consideration, and we tried to reorder the introduction to improve its logical flow and clarity. Our article presents a zirconia crown treatment for early childhood caries. This way, the introduction section starts by explaining what ECC is, and afterward we present some demographics which are necessary according to Journal’s guidelines. Later on, we briefly present the gold standard of treatment, and we conclude the introduction section by giving all-knowing information about zirconia dental crowns. We appreciate your suggestions and strive to continually improve the quality of our work.

  1. In relation to the case, it should be specified why the case can be considered of value for the discipline, what is the main contribution and because it helps the clinician who cares for this type of patient in their clinical.

This case can be considered valuable for the discipline of pediatric dentistry as it describes a successful treatment of a 4-year-old boy with high individual caries risk, severe dental anxiety and phobia, and an aesthetic demand all under general anesthesia and there are no similar cases available in the literature.

  1. The introduction should be limited only to the problem to be solved; the introduction should be synthesized.

Thank you for your feedback on the introduction section of our article. We understand your suggestion to limit the introduction to the problem to be solved, but we believe that it is necessary to also introduce the early childhood caries and the use of zirconia dental crowns. These are essential elements that cannot be omitted as they provide context and background information for the case report. Our article title "Full mouth treatment of early childhood caries with zirconia dental crowns. A case report" indicates the focus of the study and provides a clear indication of the problem being solved. However, we are open to suggestions and will consider your feedback as we strive to improve the quality of our work.

  1. One aspect to improve is the description of the figures, they are very good and adequate images; however, only 1, 2 and 3 are cited in the text and the description is very poor. I think that one way to solve this issue is to group and make scenarios like Figure 1A, 1B, 1C, etc.

Thank you for your valuable feedback on the description of the figures in our article. We understand the importance of providing adequate information to support the figures and have taken your suggestion to heart. We have reorganized the images and made sure to cite all of them in the main text to ensure that the information is clear and understandable. We appreciate your input and are committed to making continuous improvements to enhance the quality of our work.

  1. Be specific in what type of pulp treatment was used in each of the teeth.

We appreciate your feedback and understand the importance of providing specific information about the type of pulp treatment used in each tooth. We have taken your suggestion into consideration and have added this information to the main text to provide a more comprehensive understanding of the treatment performed.

  1. The discussion is practically a continuation of the introduction; must be redone. The report can be of great value because of the little information available on similar cases; however, it should be better addressed and the discussion deepened, describing what the true value of the case is, how clinicians can take advantage of this information, the diagnostic value, the implications, and what can be done at the dental care level.

In response to the feedback, we have thoroughly reworked the discussion section to provide a more comprehensive and in-depth analysis of the case and its implications for dental care. We hope that this will help clinicians better understand and apply the information presented in the case report.

  1. Discussion need more clinical interpretation. The discussion is very descriptive and does not deepen or compare with other treatment alternatives. Individual sentences are presented but not discussed or compared.

We have taken their feedback into consideration and have made efforts to deepen and add more clinical interpretation to this section.

  1. One aspect to discuss is the cost of treatment and establish in what type of practice this treatment option can be used. In the discussion describe what is expected with a long-term follow-up. Clarify and deeper the limitations identified and especially the future proposed works and recommendations based on the information generated in the article.

We have taken into consideration your comment regarding the cost of treatment and its feasibility in different practice settings. We have added information in the discussion about the treatment being carried out in a private dental clinic and the cost being covered by the patient's family. We also mentioned the importance of continuous monitoring and long-term follow-up. In terms of future works and recommendations, we emphasized the need to address the psychological and emotional aspects of pediatric dental care, investigate cost-effective options, increase patient and family involvement in oral hygiene practices, and consider cultural and linguistic barriers in patient education and communication.

  1. Normally, clinical cases have between 10 and 15 references. It is very important to limit yourself to this number and highlight the relevance of the clinical results obtained. For example, in the next sentence, is nor necessary the references “This article describes the restoration of all deciduous teeth in a 4-year-old boy with 96 high individual caries risk under general anesthesia [28,29].” It is important to check all text.

Thank you for your time and efforts in providing valuable feedback. We appreciate your insights and suggestions. We are available for any further information or clarification needed.

Reviewer 2 Report

This article presents a subject that no longer corresponds to an absolute novelty such as the treatment of deciduous dentition with esthetic zirconia crowns.

I would like to make several general remarks about it before analyzing the paper. 

The first is that I consider the treatment to be too aggressive since, according to the radiographic analysis, many of the teeth can be treated more conservatively than with full veneer crowns using the principles of minimally invasive dentistry.

On the other hand, and much more important, the article does not mention any type of preventive treatment for the child (change of habits, diet analysis, fluoridation...) which is fundamental and much more important than the treatment itself.

Even the very fact of subjecting a 4 year old child to a treatment under general anesthesia that can be addressed by other methods of behavioral control or conscious sedation with less risk to the child and that has not been described in the article the reason for not choosing it, makes me not consider the article suitable for this journal.

In the merely formal aspect, the case is well described, although it is true that aspects such as occlusion have not been very correct.

Author Response

Reviewer 2:

  1. This article presents a subject that no longer corresponds to an absolute novelty such as the treatment of deciduous dentition with esthetic zirconia crowns.

Thank you for your comment, we understand your point that the treatment of deciduous dentition with esthetic zirconia crowns is not a novel subject. Our motivation for presenting this case was the absence of similar cases in the literature, and we hope that this will add to the existing body of knowledge in this field.

I would like to make several general remarks about it before analyzing the paper. 

  1. The first is that I consider the treatment to be too aggressive since, according to the radiographic analysis, many of the teeth can be treated more conservatively than with full veneer crowns using the principles of minimally invasive dentistry.

Thank you for your review and comment. We understand that you believe the treatment was quite aggressive. We agree with your perspective, and we acknowledge that other treatment options, such as minimally invasive dentistry, should have been considered. However, in this particular case, the patient was not cooperative and had already been seen by other pediatric dentists without success. The use of general anesthesia and full zirconia crowns was ultimately chosen due to the high esthetic demands of the parents, as well as the patient's severe dental phobia. While an adhesive approach would have been preferred, it was deemed necessary to use full crowns due to the potential for failure of the composite. Nevertheless, this was a unique and individual treatment plan proposed for this specific patient, taking into account all of their individual circumstances.

  1. On the other hand, and much more important, the article does not mention any type of preventive treatment for the child (change of habits, diet analysis, fluoridation...) which is fundamental and much more important than the treatment itself.

Thank you for your valuable feedback on our article. We appreciate your insightful comment regarding the importance of preventive treatment for the child. We have taken your suggestion into consideration and have added a section to the main text to address this important aspect of pediatric dental care.

  1. Even the very fact of subjecting a 4 year old child to a treatment under general anesthesia that can be addressed by other methods of behavioral control or conscious sedation with less risk to the child and that has not been described in the article the reason for not choosing it, makes me not consider the article suitable for this journal.

 Thank you for your constructive feedback on our article. We understand your concerns about the use of general anesthesia for a 4 year old child, and we appreciate your suggestion regarding alternative methods such as behavioral control or conscious sedation.

We would like to inform you that the patient in question had already been seen by multiple pediatric practitioners and had not responded to previous attempts at treatment using various methods such as nitrous oxide, hypnosis, and medication for his dental phobia. Despite these efforts, the patient remained uncooperative, which made general anesthesia the most practical solution for both the patient, parents, and practitioners involved.

Additionally, we would like to highlight that the cost of general anesthesia in France is covered by public insurance and is more affordable than in other countries. We understand the importance of preventive treatment and it is something that we took into consideration when writing our article.We are open to considering any further suggestions or recommendations you may have and are available to provide any additional information you may need.

  1. In the merely formal aspect, the case is well described, although it is true that aspects such as occlusion have not been very correct.

We appreciate your keen attention to detail and your observations regarding the occlusion aspect. We understand that it can be challenging to achieve perfect occlusion, especially when all the teeth have been trimmed. Nevertheless, we would like to highlight that despite these difficulties, the patient has been pain-free and has been able to eat and swallow correctly since the treatment. Additionally, we have ensured that a follow-up will be conducted until the exfoliation of the primary dentition. We hope that this information will provide additional context and reassurance regarding the treatment plan and its outcomes.

Thank you again for your valuable input.

Round 2

Reviewer 1 Report

In general, the authors have responded to the suggestions made previously. Just check some typographical errors, the number of figures is incorrect and check their reference in the text.

Author Response

Thank you for your feedback. We appreciate your time and attention to detail. We have carefully reviewed and corrected any typographical errors in the manuscript. We also checked the number of figures and made the necessary adjustments to ensure that they are correct. Additionally, we have reviewed our references and made sure that they are properly cited in the text. Thank you again for your helpful suggestions.

Reviewer 2 Report

Thanks to the authors for the improvement in the article. 

However, I still do not see an overall improvement for a treatment that I believe should have been done differently and that now tries to justify retrospectively.

In my opinion it is too aggressive for a 4 year old child and the result that is described as a "functional rehabilitation" is far from having been completed with an acceptable occlusion. 

In the formal aspect the writing is good, it is in accordance with what is required for a scientific article with a few points:

In the abstract they should include the imperative need for preventive treatment for children with ECC.

The same goes for the Introduction. If prevention is not carried out in this type of patient, any treatment, and even more so an aggressive treatment such as this one, may fail, and if any of the crowns fail? Are they going to subject the child to general anesthesia again?

They speak on line 46 of index dmft. The WHO recommends the use of index dft for primary dentition.

There are more contraindications for the use of zirconium crowns and one of them is the occlusion of the patient. In the initial photos they do not present the child with occlusal photos, which they do in the final photos with a final occlusion that frankly could be improved. 

I do not consider that the aesthetic requirements of the patient or his parents in this case should condemn the child to a treatment that is functionally incorrect.

They have justified in the article much more the use of preventive strategies, which I value but I think that it should prevail much more in the article, since I consider it more important for the patient than the treatment itself. If the habits that led to that large amount of caries are not corrected, the treatments will fail and aggressive caries will recur in the permanent teeth that are yet to erupt.

Author Response

Thank you for your valuable feedback and confirm that we fully agree you’re your assessment regarding the importance of primary dental prevention. We acknowledge that the treatment itself is just a temporary measure and that long-term success depends on the patient's adoption of proper oral hygiene habits and a healthy diet. We are committed to continuing to emphasize the importance of preventive strategies with our patients, and the patient in question is currently being screened regularly in our clinic to ensure they are familiarized with proper dental care techniques and motivated to maintain good oral hygiene. Ultimately, our goal is to help our patients achieve optimal oral health and prevent the recurrence of caries in their permanent teeth.
